# Genomic epidemiology of Delta SARS-CoV-2 during transition from elimination to suppression in Aotearoa New Zealand

Lauren Jelley[1,2,6], Jordan Douglas [3,6], Xiaoyun Ren[1], David Winter[1], Andrea McNeill[1], Sue Huang [1], Nigel French [4], David Welch[2], James Hadfield[5], Joep de Ligt [1] & Jemma L. Geoghegan [1,2 ✉]

New Zealand's COVID-19 elimination strategy heavily relied on the use of genomics to inform contact tracing, linking cases to the border and to clusters during community outbreaks. In August 2021, New Zealand entered its second nationwide lockdown after the detection of a single community case with no immediately apparent epidemiological link to the border. This incursion resulted in the largest outbreak seen in New Zealand caused by the Delta Variant of Concern. Here we generated 3806 high quality SARS-CoV-2 genomes from cases reported in New Zealand between 17 August and 1 December 2021, representing 43% of reported cases. We detected wide geographical spread coupled with undetected community transmission, characterised by the apparent extinction and reappearance of genomically linked clusters. We also identified the emergence, and near replacement, of genomes possessing a 10-nucleotide frameshift deletion that caused the likely truncation of accessory protein ORF7a. By early October, New Zealand moved from an elimination strategy to a suppression strategy and the role of genomics changed markedly from being used to track and trace, towards population-level surveillance.

[1] Institute of Environmental Science and Research, Wellington, New Zealand. [2] Department of Microbiology and Immunology, University of Otago, Dunedin, New Zealand. [3] Centre for Computational Evolution, School of Computer Science, University of Auckland, Auckland, New Zealand. [4] Tāwharau Ora/ School of Veterinary Science, Massey University, Palmerston North, New Zealand. [5] Fred Hutchinson Cancer Research Centre, Seattle, Washington, USA. [6] These authors contributed equally: Lauren Jelley, Jordan Douglas. ✉email: jemma.geoghegan@otago.ac.nz

Since early 2020, the World Health Organisation has monitored the global evolution of *Severe acute respiratory coronavirus 2* (SARS-CoV-2). As new genomic variants emerged that posed an increased threat to public health, they have been classified as 'Variants of Concern' (VOC), 'Variants of Interest' (VOI) or 'Variants Under Monitoring' (VUM), and most VOCs and VOIs are designated a Greek letter. Such variants typically have one or more of an increase in transmissibility, a change in virulence, and the ability to escape pre-existing immunity compared to other genomic variants.

In December 2020, three distinct but related lineages of SARS-CoV-2 were detected in India, named subvariants B.1.617.1, B.1.671.2 and B.1.617.3[1]. From these lineages, B.1.617.2 outcompeted the others to become dominant in India and was soon after characterised as a VOC, denoted Delta. By March 2021, India experienced a significant second wave of the pandemic associated with the Delta variant. Just two months later, reported cases in India accounted for half of all global cases and quickly overwhelmed healthcare services[2]. The Delta variant rapidly spread around the world and became dominant globally during 2021, largely outgrowing Alpha (B.1.1.7) and other variants.

The Delta variant possesses at least 13 non-synonymous mutations compared to ancestral variants and its growth advantage can be explained primarily due to both immune evasion and a 40-60% increase in transmissibility compared to Alpha[3,4]. The astonishing speed at which Delta spread led to outbreaks in many countries that had previously achieved elimination. For example, Delta outbreaks in Australia, Fiji, Vietnam, Singapore and Taiwan demonstrated how previous public health measures used to control COVID-19 were less effective against this highly transmissible variant[5–9].

New Zealand eliminated community transmission of SARS-CoV-2 by mid-2020 and continued to pursue a zero-COVID policy up until late in 2021[10]. Several small but quickly controlled SARS-CoV-2 outbreaks were detected in the community after the virus was first eliminated[11]. But for the most part, cases of COVID-19 were largely restricted to managed isolation and quarantine facilities at the border, where returning New Zealanders were required to undergo at least 14 days of quarantine. These border control measures averted numerous virus incursions and, as a result, New Zealand saw an increase in life expectancy over the first two years of the pandemic[12,13].

In August 2021, following a brief period of reciprocal quarantine-free travel with Australia, a single community case was detected in Auckland, New Zealand with no immediately apparent epidemiological link to the border. New Zealand entered its second nationwide lockdown as public health officials awaited genome sequencing to identify the case's origin[14]. Genomics identified that the case was infected with the highly transmissible Delta variant, resulting in a large outbreak of 8974 reported community cases and 20 deaths between 17 August and 1 December 2021. In early October, due to the inability to eliminate Delta even under the most stringent lockdown measures, and with the aid of high vaccination rates, New Zealand moved from elimination to suppression, aimed at minimisation and protection[15,16]. Stay-at-home orders persisted in Auckland and sporadically across various regions of the country until early December. By the end of 2021 this Delta outbreak was the largest single-origin outbreak in New Zealand—the second-largest occurred in August 2020 with fewer than 200 reported community cases[17].

Here we describe a large-scale outbreak of SARS-CoV-2 in New Zealand and show how genomics was used in real-time to first help track and trace cases of COVID-19 in the community and, later, with the change to suppression, to monitor its spread and evolution. This genomic surveillance identified numerous examples of cryptic virus transmission by the apparent extinction and reappearance of genomically linked clusters. Additionally, we identified the emergence and dominance of a lineage possessing a 10-nucleotide frameshift deletion likely rendering an accessory protein, encoded by the ORF7a gene, functionally impaired, within this outbreak.

## Results

**New Zealand's Delta outbreak**. We generated 3806 high-quality genomes, designated as Delta VOC (B.1.617.2; lineage AY.39.1.1), sampled between 17 August and 1 December 2021. To inform the goal of elimination of COVID-19 in the community, between 17 August and 3 October, >88% of reported cases resulted in high-quality genomes (Fig. 1), which directly aided contact tracing efforts. The proportion of cases subject to genome sequencing fell to an average of 33% from 4 October to 1 December as a consequence of the change in strategy from elimination to suppression, as well as the increase in reported cases meaning proportionally fewer samples being referred for genomic sequencing. The number of reported cases during the Delta outbreak peaked on 16 November and case numbers have since declined markedly. Delta cases were last detected in March 2022 through routine genomic surveillance, albeit at low levels among a background of cases with the highly transmissible Omicron variant[18]. The data generated originated from 15 out of 20 District Health Boards located across the country (Fig. 1). In line with the geographic location of the majority of cases, the vast majority of genomes sampled over this time frame were from Auckland (19%), Counties Manukau (38%) and Waitematā (31%) —the three metropolitan District Health Boards that surround Auckland, New Zealand's most populous city and where the outbreak started (Fig. 1).

Much like previous COVID-19 incursions in New Zealand, the Delta outbreak was genomically linked to a single introduction into the community (Fig. 1; Supplementary Figure 1). While no definitive epidemiological link to the border could be established, genome sequencing revealed four genomes sampled from a managed isolation and quarantine facility in Auckland that were genetically indistinguishable to the consensus genome of the first reported case in the community. These cases originated from two separate travel groups who had recently returned from New South Wales, Australia in the week prior to the community outbreak. In addition, several genetically indistinguishable genomes were sampled from the New South Wales community during this time.

**Widespread undetected community transmission**. Until early October 2021, genomics was used to trace every reported case of COVID-19 in New Zealand. However, the ability of genomics to identify transmission chains became hampered due to widespread undetected community transmission. This was apparent from numerous genomic lineages that remained undetected for several weeks only to reappear again (Fig. 2). As reported case numbers rose and New Zealand moved to a suppression strategy, the focus moved from track and trace, to monitoring the spread of active genomic clades (Fig. 2). Cases reported from Auckland's metropolitan areas were evident in nearly all clades whereas cases reported from more regional localities showed much less genomic variation. As such, there was strong evidence of transmission 'seeding' from Auckland's metropolitan areas into other regions, resulting in the emergence of new, dominant clades that amplified, diverged and ultimately spread into further locations. An example is the Waikato region (purple in Fig. 2), where there was evidence of widespread community transmission of a single, dominant clade that subsequently diverged into four subclades, before a period characterised by cryptic spread and transmission to other regions.

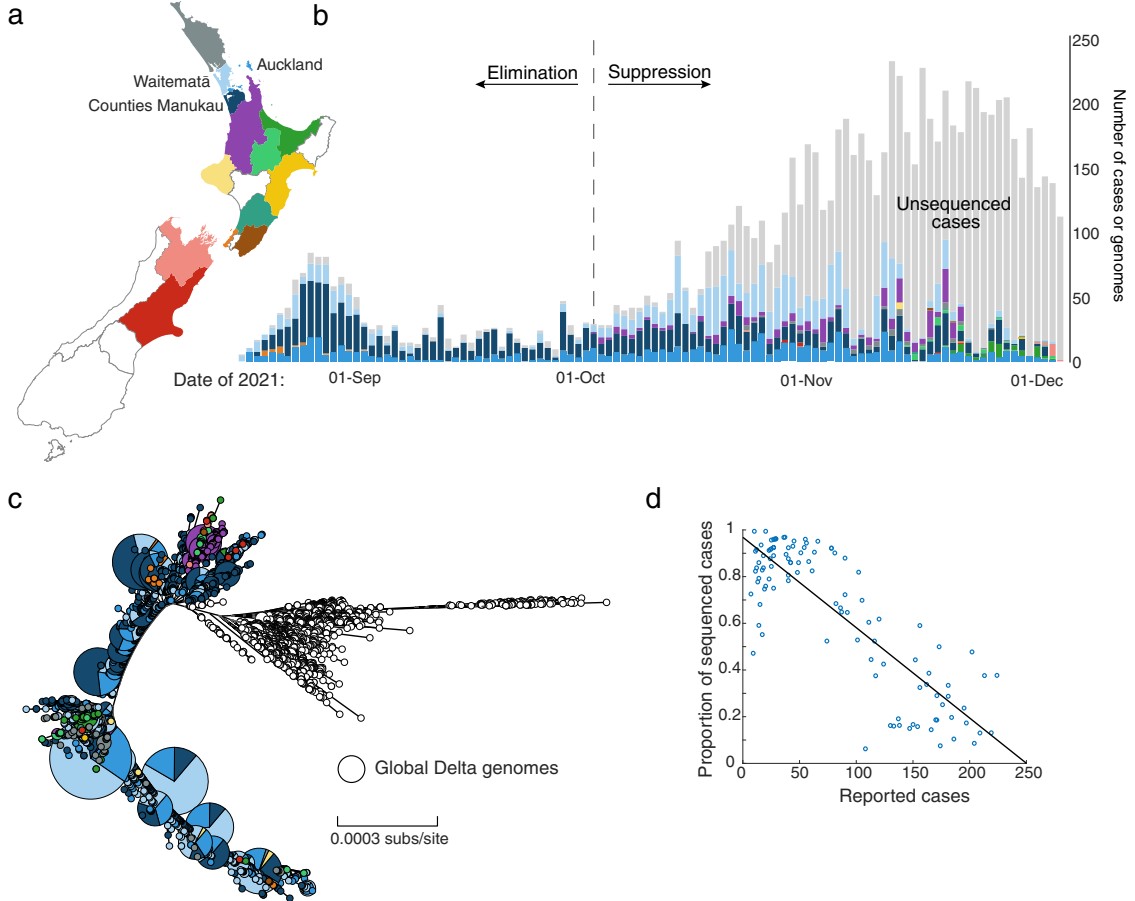

**Fig. 1 Distribution and evolution of Delta SARS-CoV-2 genomes in New Zealand. a** Map of New Zealand where regions are categorised by District Health Board. Coloured regions indicate positive COVID-19 cases were reported in that location between 17 August and 1 December. **b** The number of genomes sequenced coloured by region, corresponding to the map in **a**. **c** Maximum likelihood phylogenetic tree displayed as a GrapeTree[51] coloured by region along with global genomes (white). Branch lengths are scaled by the number of nucleotide substitutions per site. See Supplementary Figure 1 for a regular maximum likelihood phylogenetic tree. **d** Proportion of daily reported cases sequenced versus number of daily reported cases, with least-squares line.

**Frameshift deletion in ORF7a**. The outbreak was characterised by the emergence of a 10-nucleotide frameshift deletion in the ORF7a gene, which produces one of 11 accessory proteins in the SARS-CoV-2 genome (Fig. 3; see Supplementary Figure 2). This deletion spanned positions 27694 and 27704 and resulted in a predicted truncated protein of only 103 amino acids (compared to the 121 long amino acid wildtype product without the deletion), containing three altered amino acids prior to a premature stop codon. The first virus genome possessing the ORF7a deletion was sequenced on 27 August and by early October genomes with this deletion accounted for over 90% of sequenced cases (Fig. 3). In order to assess whether the difference in prevalence was confounded by epidemiological factors such as the infected person's age (i.e. younger age groups may be more likely to transmit), we compared the ages of people infected with the two lineages. We found little difference in age between cases with the deletion (mean = 31, median = 30, interquartile range = 18–43) and those without (mean = 27, median = 24, interquartile range = 14–40) thus suggesting that age was unlikely to be linked to the increase in the prevalence of truncated ORF7a genomes (Fig. 3).

The ORF7a deletion is predicted to have truncated three structural features at the C-terminus of accessory protein 7a (Fig. 3):

i. the transmembrane helix;
ii. the endoplasmic reticulum (ER) localisation signal KRKIE; and
iii. the post-translational modification site K119[19].

This deletion would likely render accessory protein 7a unable to carry out regular functions in the ER/Golgi complex. Despite this defect, viruses with the truncated ORF7a lineage clearly remained viable and indeed quickly became the dominant Delta lineage in New Zealand.

We estimated the $R_{eff}$ of both lineages through time (Fig. 4). The wildtype ORF7a lineage peaked when the first cases were discovered, with an $R_{eff}$ of 6.8 (and a 95% credible interval (CI) of 2.1–15). Following this, $R_{eff}$ declined rapidly to around 1 for the remainder of the outbreak, likely as a result of a stringent public health response and high vaccination rates (86.4% of the eligible (>12 years of age) population by 1 December). Similarly, the truncated ORF7a lineage peaked shortly after it was first detected, with a mean $R_{eff}$ of 2.1 (95% CI: 0.1–4.5). Although the truncated ORF7a lineage saw some initial success, its $R_{eff}$ gradually declined to around 1. There is no evidence that the wildtype nor truncated ORF7a lineages have any significant differences in transmissibility.

## Discussion

New Zealand's elimination strategy for controlling SARS-CoV-2 has been hailed a success, and the nation saw low levels of mortality during the first two years of the pandemic[12,13]. Genomics played an essential role in aiding the COVID-19 elimination strategy, where, until early October 2021, all positive cases were referred for sequencing and integrated with epidemiological data to directly inform control efforts. Genomics of SARS-CoV-2 helped identify

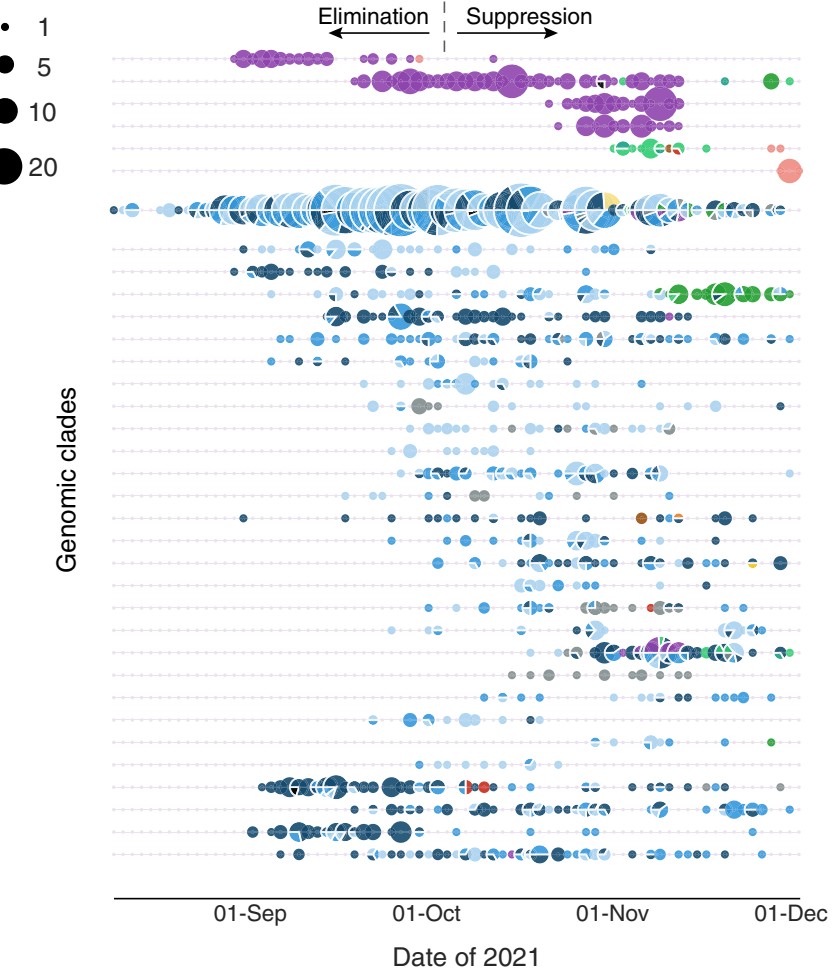

**Fig. 2 Active genomic clades during the Delta SARS-CoV-2 outbreak.** Bead plot showing a summary of active genomic clades per day between 17 August and 1 December 2021 coloured by the geographical region (District Health Board) shown in Fig. 1a. Each string of beads represents a genomic clade, and the size of each bead corresponds to the number of reported cases per day within each clade. Genomic clades were defined as involving at least one single nucleotide polymorphism (SNP) and including at least 10 descendants.

cluster membership where epidemiological links were unclear and link community cases to the border[11,17,20]. Here we show that a single incursion event, likely from a managed isolation and quarantine facility, was responsible for New Zealand's third wave of COVID-19. Although no clear epidemiological link was ever established, New Zealand's August 2021 Delta outbreak was genomically linked to cases at the border as well as to a concurrent outbreak in New South Wales, Australia.

An estimate of the rate of transmission showed that $R_{eff}$ continued to remain ~1 despite the most stringent lockdown measures[10]. This was in contrast to the effect of lockdown measures on ancestral genomic variants in New Zealand where $R_{eff}$ fell close to zero during the first wave in 2020[21,22] even following multiple superspreading events, thus enabling virus elimination. During the elimination phase of this Delta outbreak, genomic clades often appeared to go extinct only to reappear weeks later, suggestive of widespread cryptic transmission. With the number of reported cases increasing, lockdown measures unable to prevent onward transmission, and increasing vaccination coverage, New Zealand moved from an elimination strategy to a suppression strategy by early October[14,15]. The change in strategy changed the role of genomics, which shifted from one focused on contact tracing and origin detection to one targeted at the level of the New Zealand population, with a specific focus on viral evolution.

Genomic surveillance showed that the New Zealand Delta outbreak was characterised by the emergence and subsequent increase in frequency of genomes containing a 10-nucleotide frameshift deletion in ORF7a. The polymorphic nature of this deletion provided greater resolution to the epidemiological picture, enabling cases to be linked to transmission chains and therefore aiding contact tracing efforts during the elimination period among otherwise relatively clonal genomes. The ORF7a gene itself encodes accessory protein 7a, which has strong sequence similarity with its ortholog within the SARS-CoV-1 genome[23]. Accessory protein 7a is a transmembrane protein with a seven-stranded β-sandwich fold[24]. Although considered non-essential, it is involved in a range of functions including antagonisation of the host interferon type 1 response[25], binding to CD14 + monocytes in human peripheral blood[26], involvement in protein trafficking[27] and regulation of the ORF7b peptide[28], and is known to induce apoptosis when overexpressed[29,30].

Although highly variable in position and length, frameshift deletions that induce truncation of accessory protein 7a are not unique. There are numerous independent occurrences of such a mutation across both A and B lineages of SARS-CoV-2, including among VOCs[31–38]. Global genomic surveillance of SARS-CoV-2 has shown that mutations frequently occur at the downstream regions of ORF7a with such changes often leading to premature stop codons resulting in protein products with reduced functional

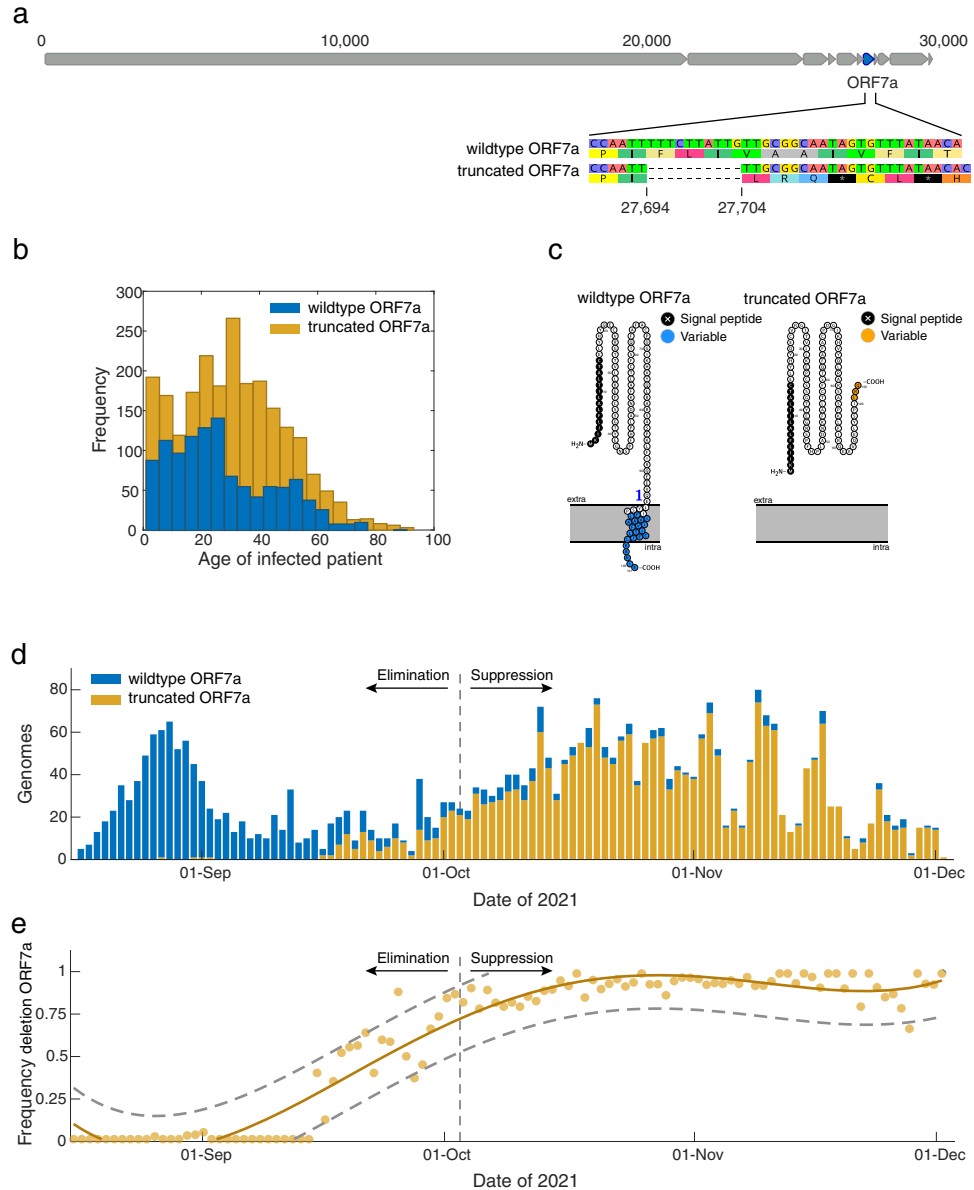

**Fig. 3 Emergence of a truncated ORF7a protein. a** Genome structure of SARS-CoV-2 illustrating the position of ORF7a along with an alignment showing the 10-nucleotide deletion between positions 27694 and 27704 and resulting frameshift mutation. **b** Age distribution of people infected with SARS-CoV-2 possessing virus genomes with (orange) and without (blue) the 10-nucleotide deletion in ORF7a. **c** Protein structure of the truncated and wildtype accessory protein 7a variants predicted by Protter[52]. A transmembrane helix is shown crossing the endoplasmic reticulum membrane for the wildtype variant and the variable amino acids are indicated. **d** Genomes sequenced through time with (orange) and without (blue) the 10-nucleotide deletion in ORF7a. **e** The frequency of genomes with the truncated ORF7a peptide (orange line), a 95% prediction interval (grey dashed lines) and the raw data (orange points) over time.

activity. Such mutations in ORF7a appear to be associated with various viral mechanisms, including reduced viral suppression of the immune response[39], an increase in viral progeny[35], and a seemingly rapid increase in viral spread[33]. Indeed, such "dispensable" accessory proteins are known to gain and lose functional activities quite rapidly in evolutionary time, across a wide range of viral taxonomies[40–42]—perhaps as a means to better explore their fitness landscapes and co-evolve with their host. This certainly appears to be the case for accessory protein 7a, which has repeatedly seen sweeping structural mutations across a range of geographical regions, while still maintaining a high level of fitness.

The truncated ORF7a lineage saw some initial success in New Zealand and quickly became the dominant variant in the community. However, this may have simply been a founder effect that coincided with a relaxation of public health restrictions. Overall, there are too many confounding factors and an insignificant difference in reproduction number to determine whether the truncated variant had any enhanced transmissibility over the wildtype.

Overall, genomics has played a key role in New Zealand's science-led response to the SARS-CoV-2 pandemic[43]. For example, this team provided genomic data and analysis that informed the public health response in real-time. The Delta outbreak not only changed New Zealand's strategy for controlling SARS-CoV-2, but it also changed the role of genomics. That is, to aid in eliminating the virus, genomic sequencing assisted public health officials by directly linking the source of the outbreak to the border, providing assurance that the outbreak was linked to a single origin, as well as assisting with real-time contact tracing to

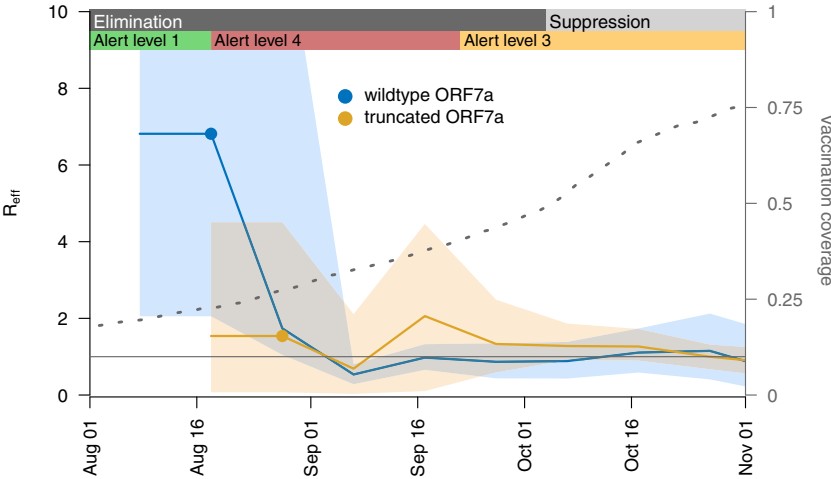

**Fig. 4 Estimates of the effective reproduction number ($R_{eff}$) over time.** Mean estimate of the effective reproduction number ($R_{eff}$) over time (solid line) and its 95% credible interval (shaded area), based on genomes with the truncated (orange) or wildtype (blue) ORF7a. One $R_{eff}$ estimate is made per 10-day period following detection of the first community case (filled circle) for either lineage, plus one additional estimate between this period and the first transmission event (i.e. the root of the tree). The right axis shows the proportion of the eligible population (12+ years old) who had received two doses of the Pfizer-BioNTech COVID-19 vaccine. The nationwide change from elimination to suppression strategy is indicated, as are regional changes between mitigation settings in Auckland (the epicentre of the outbreak), with 'Alert level 4' being the most stringent[10].

assign cases to clusters. This genomics-informed contact tracing usefully highlighted widespread cryptic transmission, informing decision makers about the trajectory of the outbreak. Now, however, much like other parts of the world where the focus has shifted to monitoring genomic variants, genomic surveillance of SARS-CoV-2 continues to inform public health responses in New Zealand, but at a different granularity.

## Methods

**Ethics statement**. Nasopharyngeal samples that had positive results for SARS-CoV-2 by real-time reverse transcription PCR were obtained from medical diagnostic laboratories located throughout New Zealand. Under contract for the New Zealand Ministry of Health, the Institute of Environmental Science and Research has the approval to conduct genomic sequencing and phylogenetic analysis for surveillance of notifiable diseases.

**Genomic sequencing of SARS-CoV-2**. For cases reported between 17 August and 1 December 2021 a random proportion of COVID-19 community cases were referred to the Institute of Environmental Science and Research, New Zealand for genome sequencing. In brief, viral extracts were prepared from respiratory tract samples in which SARS-CoV-2 was detected by rRT-PCR. Extracted RNA was subjected to whole-genome sequencing using the Oxford Nanopore Technologies R9.4 chemistry by following the Midnight protocol v6[44], which contains a 1200-bp primer set tiling the SARS-CoV-2 genome. Consensus genomes were generated through a standardised pipeline (https://github.com/ESR-NZ/NZ_SARS-CoV-2_genomics) based on the original ARTIC bioinformatics pipeline (https://artic.network/ncov-2019/ncov2019-bioinformatics-sop.html).

**Phylogenetic analysis**. A total of 3806 high-quality genomes were generated from New Zealand's 2021 Delta outbreak sampled between 17 August and 1 December. Genomes were assigned to the B.1.617.2 lineage using PANGO (v.3.1.20)[1] (https://github.com/cov-lineages/pangolin/releases/tag/v3.1.20). Four genomes sequenced from New Zealand's managed isolation and quarantine facility were also included that were genetically identical to the consensus sequences of the first reported community case and were sampled the week prior to the outbreak. These genomes were aligned along with 1457 global genomes uniformly selected at random from GISAID[45] sampled over the same time, including four genomes sequenced in New South Wales, Australia that were genetically identical to the presumed index cases in New Zealand and were sampled 10 days prior to the first reported community case (GISAID accessions: EPI_ISL_3426406, EPI_ISL_3426050, EPI_ISL_3426425 and EPI_ISL_3426179).

Genomes were aligned using Nextalign[46] with Wuhan-Hu-1 (NC_045512.2) used as a reference. A maximum-likelihood phylogenetic tree was estimated using IQ-TREE (v 1.6.8)[47] using the Hasegawa-Kishino-Yano (HKY + Γ)[48] nucleotide substitution model with a gamma distributed rate variation among sites (the best fit model was determined by ModelFinder[49]), and branch support was assessed using

the ultrafast bootstrap method[50]. The resulting phylogenetic tree was visualised and displayed using GrapeTree[51] and annotated by the location (i.e. District Health Board) from where genomes were sampled.

To better understand the diversity and regional spread of active lineages, a 'bead' plot was generated where each string of beads represented a genomic clade, which were identified during the outbreak in real-time as mutations arose in the phylogenetic tree. Genomic clades were defined as involving at least one single nucleotide polymorphism (SNP) and including at least 10 descendants.

**ORF7a deletion and protein characterisation**. Genomes sampled between 17 August and 1 December 2021 were annotated based on whether they possessed a 10-nucleotide deletion within the Open Reading Frame (ORF) 7a between positions 27694 and 27704. This resulted in a total of 2309 and 1126 genomes with and without the ORF7a deletion, respectively. The remaining 371 consensus genomes possessed ambiguous nucleotides within this region and were therefore excluded from further analysis. The transmembrane properties of ORF7a were modelled and annotated using Protter[52].

**Estimating the effective reproduction number**. To estimate the effective reproduction number ($R_{eff}$) through time for genomes with and without the ORF7a deletion, a Bayesian birth-death skyline model[53] was implemented in BEAST 2.5[54]. First, genomes were aligned as above and subsampled down to 33% ($n = 374$ for wildtype and $n = 770$ for truncated) to facilitate convergence of the Markov chain Monte Carlo (MCMC) algorithm. $R_{eff}$ was estimated for the period between the root of the tree and detection of the first community case, and then every 10-day period up until 1 December 2021 (but due to high uncertainty in $R_{eff}$ approaching the most recent sample, we only report these estimates up until 1 November). Consecutive estimates were smoothed using an Ornstein-Uhlenbeck smoothing prior. Nucleotide evolution was modelled using the HKY substitution model with frequencies estimated. MCMC chains were run until the effective sample size of reported parameters exceeded 200. To help speed up MCMC convergence, we used the efficient tree operators in the BICEPS package[54] and adaptable operator weighting[55].

**Reporting summary**. Further information on research design is available in the Nature Research Reporting Summary linked to this article.

## Data availability
Genomic data generated here are available on GenBank under accession numbers detailed in Supplementary Data 1, along with the patient age and region from where samples were collected.

## Code availability
All software used here is open source and parameters clearly described in the Methods section.

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

## Acknowledgements

Genome sequencing of SARS-CoV-2 was funded by the Ministry of Health of New Zealand. We thank the diagnostic laboratories that performed the initial RT-PCRs and referred samples for sequencing as well as the public health units for providing epidemiological data. We thank all those who have contributed SARS-CoV-2 sequences to GISAID. J.L.G. is funded by a New Zealand Royal Society Rutherford Discovery Fellowship (RDF-20-UOO-007) and L.J. is supported by a University of Otago Doctoral Scholarship.

## Author contributions

J.L.G. developed the concept; J.d.L., X.R. and D.W. performed the genomic sequencing; J.L.G., L.J. and J.D. performed the data analysis; J.L.G., L.J. and J.D. wrote the initial draft; L.J., J.D., X.R., D.W., A.M., S.H., N.F., D.W., J.H., J.d.L. and J.L.G. contributed to review and editing the final version; J.L.G. provided management and oversight for the project.

## Competing interests

The authors declare no competing interests.
