## [Peer Review File · Nature Communications]

Genomic epidemiology of Delta SARS-CoV-2 during transition from elimination to suppression in Aotearoa New ZealandREVIEWER COMMENTS

Reviewer #1 (Remarks to the Author):

This is an interesting short paper about the New Zealand SARS-CoV-2 genomic epidemiology response. I think the NZ response has been exemplary, and I fully believe in the power of genomic epidemiology to help public health response to epidemics. However, I think this paper would be much improved with additional methodological and background information related to some of the primary results, and I think it leaves out important details about how these specific genomic epidemiological analyses helped the specific public health response in NZ.

Comments/suggestions.

1. The bead plot in Figure 2 is an intuitive visual approach to better understand the size and duration of transmission chains. But as far as I can tell there is no description of how the bead plot was made, other than in the Figure legend itself ("Each string of beads represents a genomic clade and the size of each bead corresponds to the number of reported cases per day within each clade. Genomic clades were defined as involving at least one single nucleotide polymorphism (SNP) and including at least 10 descendants.") But how were the clades identified as separate clades? The visual description of phylogenetic clades as transmission chains/strings is nice, but it necessarily compresses phylogenetic branching events and makes some external nodes appear to be internal beads on the string, so the question is how did the authors decide how to separate phylogenetic clades into genomic clades / strings? This NZ outbreak was all due to a single introduction, which makes it harder (for me) to understand how separate clades were identified for this plot.
2. I think the GrapeTree in Figure 1.C. is a bit confusing, as the pie charts completely obscure the phylogenetic information. All that we end up seeing is a string of pie charts overlaid on a phylogeny with presumably many polytomies? Could you add in a regular phylogeny as well?
3. I've never seen a Streamgraph and I find that Figure 1.B. is also quite confusing.
4. On line 197 the authors state that, "There was strong evidence of transmission "seeding" from Auckland's metropolitan areas into other regions, resulting in the emergence of new, dominant clades that amplified, diverged and ultimately spread into further locations." But other than looking at the Figures (the phylogeny and the bead plot), the authors don't show any data or discuss what this strong evidence was. What is strong evidence? Is this seeding quantified in any way? Is this surprising or what one would expect, given the travel and quarantine restrictions?
5. On line 210 the authors state that, "There was little difference in the age distribution in cases with and without the deletion..." Is this also an eyeball method determination? Or is there a distribution comparison statistical test employed but not mentioned? Why did the authors choose to look at age distribution, of all the potential demographic or behavioral metadata that may have been linked to each sequence? Was there a hypothesis about the importance of age and SARS-CoV-2 transmission? Or of this variant specifically?
6. Figure 4, and the estimates of R_{eff} , suggest, not surprisingly, that there is such little diversity in SARS-CoV-2 that there are some instances where R_{eff} estimates are not useful. The 95% credible intervals around the wild type, until early Sept 2021, and the truncated variant, up until Oct 16, 2021, are quite wide. How useful are these estimates to any real-time public health action?
7. On line 303, the authors state that, "Overall, genomics has played a key role in NZ's science-led response...". I would be very interested in learning more about how these specific results were considered in the public health decision making process. As is, this paper is a tantalizing appetizer that suggests a large and maybe key role, but it does not go into details or provide specific results that changed decisions. The genomics did, apparently, help link cases together and to inferred importations... but was that retrospective analysis that was interesting, or was it a key piece of

information that led to a specific response? What response? On line 272 the authors state that after positive cases increased, the role of genomics changed to a focus on viral evolution.

Reviewer #2 (Remarks to the Author):

In this article, Jelley and colleagues reports the results of antionwide genomic surveillance in Aotearoa New Zealand in the midst of its then largest SARS-CoV-2 outbreak (caused by Delta) and the transition from an elimination to a suppression strategy. They provide evidence that this large outbreak resulted from a single introduction event. They report intense local circulation in the area around Auckland and relatively rare transmission towards the rest of Aotearoa New Zealand, with a presumably high level of cryptic spread. They also document the rise of a variant characterized by a truncated and most likely defective ORF7a.

I think that this study is well executed and written. There is always the question of the relative importance of SARS-CoV-2 genomic surveillance studies - there are so many out there. I consider that this one is an important study in that Aotearoa New Zealand strategy has been under such global scrutiny. The particular moment captured by this study, where the country transitioned from elimination to suppression will probably become a textbook example for zero-disease X strategies. These strategies, when well understood (like it was in ANZ), aim to a well managed transition between both phases. I am very supportive of the publication of this article in Nat Commun.

However, I think it could probably be reinforced by:

- i. the authors suggest a sizeable fraction of undetected cases. It would be interesting to use this genomic surveillance data to reconstruct incidence/case detection ratio (eg with the methods developed here:<https://www.nature.com/articles/s41467-021-26267-y>)
- ii. the authors could also try to provide estimates of migration rates/counts of migration events between regions (and in particular Auckland vs rest of ANZ), eg using migration models in TreeTime or phylogeographic diffusion models in BEAST. To a colorblind reviewer it seems that migrations to non-Auckland were more frequent after transitioning to suppression, and this despite the fact that the city was still under lockdown.

I also wonder whether the authors could not add some information about the reasons that pushed the government to start suppression. It looks like it was essentially linked to achieving control and vaccination targets (the virus with Reff around 1 and 50% vaccinated) but it would be interesting to know the details, and also whether genomic surveillance data per se was accounted for and how in the decision making process.

A minor remark:

L59: the authors write that Delta displaced Alpha. I always read that as competitive exclusion, which we have now seen can play a role (with Omicron's induced -partial- cross-immunity to Delta likely contributing to pushing it out of the picture). I am much less convinced about this mechanism having played any significant role when Delta became dominant and Alpha slowly disappeared last year in many parts of the world. At least in those places where mass vaccination had started (Europe, North America), it progressed much faster than Delta, which became dominant at a time when case numbers were really low - I d rather interpret that as Alpha having been beaten by vaccination and Delta having fared better with respect to this hurdle. Anyway, I would suggest to replace "displacing" w/ "outgrowing" (sorry for the long side note).

Best regards,

Sebastien Calvignac-Spencer

Reviewer #3 (Remarks to the Author):

New Zealand has had a unique COVID-19 experience mainly for geographical reasons and decisions around strict border controls. This manuscript is noteworthy as it describes the transition in the genomic sequencing strategy during transition from COVID zero to suppression, and provides a concise and informative description of the Delta outbreak at the time.

The methods are robust and the manuscript well written. The results are interesting, and add to the significant literature in the field describing the genomic epidemiology of various SARS-CoV-2 outbreaks globally.

Specific comments that should be considered:

Line 176/177 – not completely clear how fig 1 shows that this outbreak was linked to a single introduction. While figure is generally very informative, 1C is hard to decipher

Line 192 - This was hampered by known genomic clades going unsampled before reappearing weeks later – its not clear where this is demonstrated in the results, please clarify

The detection of the ORF7a truncation is interesting, but doesn't appear to have had an impact of transmission dynamics. Could the authors comment if similar deletions / deletions in the same region have been detected in international data? While this is addressed to some extent in the discussion, a more thorough interrogation of global data might be worthwhile.

Was the cryptic transmission detected though genomics influential in the decision to move away from elimination? This would be worth highlighting.

As New Zealand moved from elimination to suppression, what was the strategy for selection / proportion of positive case sequencing? How do these results inform an appropriate selection strategy moving forward?

We thank the reviewers for providing helpful and constructive feedback on a previous version of this manuscript. We have now revised the manuscript according to this feedback and provide a point-by-point response to each comment below:

Reviewer 1

This is an interesting short paper about the New Zealand SARS-CoV-2 genomic epidemiology response. I think the NZ response has been exemplary, and I fully believe in the power of genomic epidemiology to help public health response to epidemics. However, I think this paper would be much improved with additional methodological and background information related to some of the primary results, and I think it leaves out important details about how these specific genomic epidemiological analyses helped the specific public health response in NZ.

1. The bead plot in Figure 2 is an intuitive visual approach to better understand the size and duration of transmission chains. But as far as I can tell there is no description of how the bead plot was made, other than in the Figure legend itself ("Each string of beads represents a genomic clade and the size of each bead corresponds to the number of reported cases per day within each clade. Genomic clades were defined as involving at least one single nucleotide polymorphism (SNP) and including at least 10 descendants.") But how were the clades identified as separate clades? The visual description of phylogenetic clades as transmission chains/strings is nice, but it necessarily compresses phylogenetic branching events and makes some external nodes appear to be internal beads on the string, so the question is how did the authors decide how to separate phylogenetic clades into genomic clades / strings? This NZ outbreak was all due to a single introduction, which makes it harder (for me) to understand how separate clades were identified for this plot.

Response: The reviewer is correct that this approach necessarily compresses phylogenetic branching events – we found that traditional representations of trees were not correctly interpreted by decision makers (e.g. interpreting vertical closeness on rectangular trees as genomic similarity); we felt this visualisation conveyed the salient concepts of the branching process to those decision makers. Clades were identified during the outbreak in real-time as mutations arose in the (single-introduction) tree, with the ad-hoc requirement that a clade defined at least 10 tips at the time of classification. Thus, clades began as monophylies and later (often) became paraphylies as descendent clades were identified. Note also that large comb structures appeared (within clades), sometimes with over 100 tips, and we were thus unable to separate these out into clades. This classification scheme was necessarily imperfect but was a useful tool in communicating the spread of the virus to people not well versed in phylogenetics. Finally, we wish to stress that these clades were chosen during our

regular analysis during the outbreak and are not retrospective. We have added a description of the methods used to construct the bead plot.

2. I think the GrapeTree in Figure 1.C. is a bit confusing, as the pie charts completely obscure the phylogenetic information. All that we end up seeing is a string of pie charts overlaid on a phylogeny with presumably many polytomies? Could you add in a regular phylogeny as well?

Response: We thank the reviewer for this suggestion and we have now added a regular phylogeny in Supplementary Figure 1. The reviewer is correct in that there are a lot of polytomies and so many of the tips are obscured. For this reason we feel that the GrapeTree is visually clearer so we have maintained this in the main figure.

3. I've never seen a Streamgraph and I find that Figure 1.B. is also quite confusing.

Response: We have now revised the Streamgraph and instead plot these data as a bar graph for clarity.

4. On line 197 the authors state that, "There was strong evidence of transmission "seeding" from Auckland's metropolitan areas into other regions, resulting in the emergence of new, dominant clades that amplified, diverged and ultimately spread into further locations." But other than looking at the Figures (the phylogeny and the bead plot), the authors don't show any data or discuss what this strong evidence was. What is strong evidence? Is this seeding quantified in any way? Is this surprising or what one would expect, given the travel and quarantine restrictions?

Response: We thank the reviewer for pointing this out. We have now included Supplementary Figure 1 (see above), showing more clearly a single introduction against a background of global genomes. We have now added to the text and discussed this evidence further, including additional comments on the regional spread into other localities. During the time period that this paper encompasses, New Zealand had strict travel and quarantine restrictions in place and, consequently, the vast majority of outbreaks in New Zealand since COVID-19's elimination were due to a single introduction. We have now made this more clear in the manuscript.

5. On line 210 the authors state that, "There was little difference in the age distribution in cases with and without the deletion..." Is this also an eyeball method determination? Or is there a distribution comparison statistical test employed but not mentioned? Why did the authors choose to look at age distribution, of all the potential demographic or behavioral metadata that may have been linked to each sequence? Was

there a hypothesis about the importance of age and SARS-CoV-2 transmission? Or of this variant specifically?

Response: *Age distribution data was readily available with corresponding genomes. We hypothesised that if there was a difference in age distribution between genomes with the truncated ORF7a mutation and those without, there would likely be a confounding epidemiological reason for its increase in frequency (i.e. younger age group more likely to transmit). Although we did employ various statistical methods to compare the means and distributions initially, in the face of large discrepancies in cohort sizes ($n = 1126$ vs. $n = 2309$) we decided that a direct comparison of means is more readily interpreted than a statistical approach. We have also added median and interquartile ranges for both as well as further clarification on our motivation in the Results section.*

6. Figure 4, and the estimates of R_{eff} , suggest, not surprisingly, that there is such little diversity in SARS-CoV-2 that there are some instances where R_{eff} estimates are not useful. The 95% credible intervals around the wild type, until early Sept 2021, and the truncated variant, up until Oct 16, 2021, are quite wide. How useful are these estimates to any real-time public health action?

Response: *We agree that these HPD intervals are very broad, and in many cases are not as useful for informing a real-time response as standard epidemiological estimates of R_{eff} would be. However, for the purposes of this article, we believe that these estimates are useful for characterising the two Delta variants in the community.*

7. On line 303, the authors state that, "Overall, genomics has played a key role in NZ's science-led response...". I would be very interested in learning more about how these specific results were considered in the public health decision making process. As is, this paper is a tantalizing appetizer that suggests a large and maybe key role, but it does not go into details or provide specific results that changed decisions. The genomics did, apparently, help link cases together and to inferred importations... but was that retrospective analysis that was interesting, or was it a key piece of information that led to a specific response? What response? On line 272 the authors state that after positive cases increased, the role of genomics changed to a focus on viral evolution.

Response: *We have added further details to the concluding paragraph to make it explicit that these analyses informed public health response. We have also already published a number of examples on how genomics has played a vital role in informing the pandemic response in real-time in New Zealand (e.g. whether to place the country into lockdown, changes to national and regional alert level settings, etc.) and these studies have been cited. In the introduction, we have now cited a press conference, where the Government awaited genome*

sequencing results to establish the outbreak origin before making further decisions regarding lockdown measures, as an example of how genomic sequencing informed policy.

Reviewer 2

In this article, Jelley and colleagues reports the results of nationwide genomic surveillance in Aotearoa New Zealand in the midst of its then largest SARS-CoV-2 outbreak (caused by Delta) and the transition from an elimination to a suppression strategy. They provide evidence that this large outbreak resulted from a single introduction event. They report intense local circulation in the area around Auckland and relatively rare transmission towards the rest of Aotearoa New Zealand, with a presumably high level of cryptic spread. They also document the rise of a variant characterized by a truncated and most likely defective ORF7a.

I think that this study is well executed and written. There is always the question of the relative importance of SARS-CoV-2 genomic surveillance studies - there are so many out there. I consider that this one is an important study in that Aotearoa New Zealand strategy has been under such global scrutiny. The particular moment captured by this study, where the country transitioned from elimination to suppression will probably become a textbook example for zero-disease X strategies. These strategies, when well understood (like it was in ANZ), aim to a well managed transition between both phases. I am very supportive of the publication of this article in Nat Commun.

However, I think it could probably be reinforced by:

- i. the authors suggest a sizeable fraction of undetected cases. It would be interesting to use this genomic surveillance data to reconstruct incidence/case detection ratio (eg with the methods developed here: <https://www.nature.com/articles/s41467-021-26267-y>)
- ii. the authors could also try to provide estimates of migration rates/counts of migration events between regions (and in particular Auckland vs rest of ANZ), eg using migration models in TreeTime or phylogeographic diffusion models in BEAST. To a colorblind reviewer it seems that migrations to non-Auckland were more frequent after transitioning to suppression, and this despite the fact that the city was still under lockdown.

Response: *We agree that these would be interesting analyses that could provide great insight into the outbreak and thank the reviewer for this suggestion. However, we believe that these analyses are beyond the scope of the current article. This article is largely focused on how the New Zealand Government was informed by genomics, and these analyses, although potentially useful, were not part of our workflow.*

I also wonder whether the authors could not add some information about the reasons that pushed the government to start suppression. It looks like it was essentially linked to achieving control and vaccination targets (the virus with R_{eff} around 1 and 50% vaccinated) but it would be interesting to know the details, and also whether genomic surveillance data per se was accounted for and how in the decision making process.

Response: We now point out that, along with other factors such as high vaccination rates and increasing daily case numbers, genomics also played a key role in the decision making around control strategy.

A minor remark:

L59: the authors write that Delta displaced Alpha. I always read that as competitive exclusion, which we have now seen can play a role (with Omicron's induced -partial- cross-immunity to Delta likely contributing to pushing it out of the picture). I am much less convinced about this mechanism having played any significant role when Delta became dominant and Alpha slowly disappeared last year in many parts of the world. At least in those places where mass vaccination had started (Europe, North America), it progressed much faster than Delta, which became dominant at a time when case numbers were really low - I'd rather interpret that as Alpha having been beaten by vaccination and Delta having fared better with respect to this hurdle. Anyway, I would suggest to replace "displacing" w/ "outgrowing" (sorry for the long side note).

Response: We have replaced 'displacing' with 'outgrowing'.

Reviewer 3

New Zealand has had a unique COVID-19 experience mainly for geographical reasons and decisions around strict border controls. This manuscript is noteworthy as it describes the transition in the genomic sequencing strategy during transition from COVID zero to suppression, and provides a concise and informative description of the Delta outbreak at the time. The methods are robust and the manuscript well written. The results are interesting, and add to the significant literature in the field describing the genomic epidemiology of various SARS-CoV-2 outbreaks globally. Specific comments that should be considered:

Line 176/177 – not completely clear how fig 1 shows that this outbreak was linked to a single introduction. While figure is generally very informative, 1C is hard to decipher

Response: We thank the reviewer for pointing this out and have now included a regular phylogenetic tree in Supplementary Figure 1 that perhaps more clearly shows a single introduction against a background of global Delta genomes. We have retained the GrapeTree in the main text because so many polytomies meant many

overlapping and unclear tips.

Line 192 - This was hampered by known genomic clades going unsampled before reappearing weeks later – its not clear where this is demonstrated in the results, please clarify

Response: *We have revised this sentence to add further clarity.*

The detection of the ORF7a truncation is interesting, but doesn't appear to have had an impact of transmission dynamics. Could the authors comment if similar deletions / deletions in the same region have been detected in international data? While this is addressed to some extent in the discussion, a more thorough interrogation of global data might be worthwhile.

Response: *While we agree with the reviewer that a more thorough interrogation of published data would be ideal (and we previously attempted to do so), it would be impossible to determine whether any such deletion event is indeed real since we do not have access to raw sequencing reads and gaps could be associated with low sequencing coverage rather than a true deletion. Instead we have provided a substantial discussion of published studies that have verified that deletions in ORF7a are legitimate, as we have done so in our paper.*

Was the cryptic transmission detected though genomics influential in the decision to move away from elimination? This would be worth highlighting.

Response: *We now point out that, along with other factors such as high vaccination rates and increasing daily case numbers, genomics also played a key role in the decision making around control strategy.*

As New Zealand moved from elimination to suppression, what was the strategy for selection / proportion of positive case sequencing? How do these results inform an appropriate selection strategy moving forward?

Response: *All testing labs were encouraged to send positive samples for genomic sequencing. When the number of samples referred for sequencing out-stripped the sequencing facility's ability to process samples priority was given to cases with no known epidemiological link to other cases. Additionally, samples were selected to have broad geographic sampling both at the regional (DHB) and local (city postcode) level. Finally, samples with very high CT (>30) were not routinely sequenced at this time. During the peak of the outbreak, the rate limiting step was the number of samples that could be processed and sent to the sequencing facility in a timely manner. While not entirely random, the samples received represented a good spatial distribution*

reflecting the number of cases in each region. In addition, PCR was the only test available during this outbreak and so we believe the number of reported cases somewhat accurately reflected the number of infections.

REVIEWERS' COMMENTS

Reviewer #1 (Remarks to the Author):

The authors have adequately addressed my questions, and I support publication of this revised manuscript. Very nice work.